# Feeding Preference of Crapemyrtle Bark Scale (*Acanthococcus lagerstroemiae*) on Different Species

**DOI:** 10.3390/insects11070399

**Published:** 2020-06-28

**Authors:** Runshi Xie, Bin Wu, Haijie Dou, Cuiyu Liu, Gary W. Knox, Hongmin Qin, Mengmeng Gu

**Affiliations:** 1Department of Horticultural Sciences, Texas A&M University, College Station, TX 77843, USA; fushe001@tamu.edu (R.X.); bin.wu@tamu.edu (B.W.); haijiedou@outlook.com (H.D.); 2Department of Science & Technology Development, Beijing Industrial Technology Research Institute, No.5, Xingguang 4th Ave, Tongzhou District, Beijing 101111, China; 3College of Forestry, Nanjing Forestry University, 159 Longpan Rd., Nanjing 210037, China; ankar_liu@163.com; 4Department of Environmental Horticulture, University of Florida/IFAS North Florida Research and Education Center, Quincy, FL 32351, USA; gwknox@ufl.edu; 5Department of Biology, Texas A&M University, College Station, TX 77840, USA; 6Department of Horticultural Sciences, Texas A&M AgriLife Extension Service, College Station, TX 77843, USA

**Keywords:** *Lagerstroemia*, host range, pomegranates, apple, scale insect, exotic pest

## Abstract

Crapemyrtle bark scale (CMBS; *Acanthococcus lagerstroemiae*) is an exotic pest species that causes aesthetic and economic damage to crapemyrtles and poses potential threats to other horticultural crops in the United States. Although previous studies reported the infestation of CMBS on several alternative hosts across multiple families in Asia, its potential threats to other documented alternative hosts remain elusive and yet to be confirmed. In this study, feeding preference studies of CMBS were conducted on forty-nine plant species and cultivars in 2016 and 2019, in order to gain insight into the expansion of CMBS distribution in the United States, as well as other regions of the world. The infestations of CMBS were confirmed on apple (*Malus domestica*), *Chaenomeles speciosa*, *Disopyros rhombifolia*, *Heimia salicifolia, Lagerstroemia* ‘Spiced Plum’, *M. angustifolia*, and twelve out of thirty-five pomegranate cultivars. However, the levels of CMBS infestation on these test plant hosts in this study is very low compared to *Lagerstroemia*, and may not cause significant damage. No sign of CMBS infestation was observed on *Rubus* ‘Arapaho’, *R.* ‘Navaho’, *R. idaeus* ‘Dorman Red’, *R. fruticosus*, *B. microphylla* var. *koreana* × *B. sempervirens*, *B. harlandii*, or *D. virginiana*.

## 1. Introduction

Crapemyrtle bark scale (CMBS; *Acanthococcus lagerstroemiae* Borchsenius, 1960) originates from East Asia, and has been reported to infest important horticultural crops, such as crapemyrtles and pomegranates, in Japan, Korea, India, and China [1,2]. Since its first discovery in the United States in 2004 in Richardson, TX, the infestation of this scale insect has been confirmed in Alabama, Arkansas, Georgia, Kansas, Louisiana, Maryland, Mississippi, New Mexico, North Carolina, Oklahoma, Tennessee, Virginia [3], and Washington [4]. The most common host of CMBS, crapemyrtles (*Lagerstroemia* spp.), is an important landscape tree in the southern United States, which generates a combined market value (wholesale) of around USD 67 million per year [5]. The wide use of crapemyrtle could be due to its various growth patterns and plant architectures, flower color and duration, attractive bark features, as well as disease and pest tolerance [6]. However, infestation by CMBS can greatly affect the performance of crapemyrtles, causing aesthetic and economic damage. In addition, the widespread distribution of this exotic pest species also poses threats to other potential alternative host crops, especially native plant species such as American beautyberry (*Callicarpa americana*) in the United States [4].

Crapemyrtle bark scale is a phloem-feeding hemipteran. The development of CMBS is categorized as of incomplete metamorphosis, with a life cycle of 56 to 83 days [7]. There are between two to four generations of CMBS per year, depending on the climate zones [2,8]. In the United States, up to four generations were observed in the field in Dallas, TX [2]. Under greenhouse conditions, the development of males involves five nymphal stages, but only three for females [9]. The relatively small size of CMBS (ranging from 0.3 mm to 3 mm, depending on the developmental stages) renders difficulties for detection of the insect in the early infestation.

The damage caused by CMBS includes slow and weakened plant growth, leaf abscission, absence of flowering, death of branches, and, in severe cases, the death of the juvenile seedling or young crapemyrtle trees [10]. Another major issue with CMBS infestation is the accumulation of black sooty mold, which is caused by the honeydew secretion when CMBS nymphs actively feed on the plant [2,11]. Common treatments to control CMBS include the application of horticultural oil on the bark surface and root injection or drench of systemic insecticides [8,12]; however, the lack of information on insect biology of CMBS and the population dynamics related to its adaptation in the United States is an obstacle in managing this pest insect, as it continues to spread among southeastern United States [13]. No host resistance to CMBS has been reported among *Lagerstroemia* species and cultivars, though different levels of infestations of CMBS can often be observed in the field, which can be caused by many environmental factors, such as temperature, shade, humidity, and the presence of natural enemies [11,14,15].

Abiotic and biotic factors are known to have impacts on insect biology in terms of altering development duration and sex ratio. For example, green pea aphid (*Myzus persicae*) showed varied development durations under different temperature conditions [16]. Similarly, altered development duration and sex ratio were observed on parasitoid *Campoletis chlorideae* at different temperatures [17]. On the other hand, the development of CMBS could be greatly affected by the direct interaction between the insect and plant host, including the CMBS feeding preference and the potential insect tolerance from the plant [18], which determines the severity of infestation.

Alternative host selection and feeding preference, as important aspects of insect biology, are valuable information in terms of understanding insect population dynamics, and how the insect disperses geographically [19,20,21]. While many insect herbivores are considered specialists, others are known to maintain the ability to feed on a wide range of plant hosts, or switch hosts, as an adaptive strategy when facing adverse environments [22,23]. For example, hemipteran *Thaumastocoris peregrinus* specialized in infesting host plants within *Eucalyptus* [24]. Although most species of aphids (Aphididae) are known to be specialized on one certain host plant [25], many are still capable of feeding on multiple plant hosts [26,27,28]. Economically important aphid species, including *Aphis fabae* [29], *Aphis nasturtii* [30], *Aphis aurantia* [31], *Aulacorthum solani* [32], *Brachycaudus helichrysi* [33], *Macrosiphum euphorbiae* [34], *Myzus ornatus* [35], *Myzus persicae* [36] are considered polyphagous. Some members of Aphididae further exhibit complex life histories involving alternation of feeding between unrelated plant taxa [37]. Thus, alternative hosts remain important for the survival and perpetuation of many insect species [38,39].

In the case of CMBS, previous reports from Asia have indicated that its host range includes apple (*Malus pumila*), axlewood (*Anogeissus latifolia*), border privet (*Ligustrum obtusifolium*), boxwood (*Buxus microphylla*), Chinese hackberry (*Celtis sinensis*), Chinese-quince (*Pseudocydonia sinensis*), common fig (*Ficus carica*), food wrapper plant (*Mallotus japonicus*), Indian rosewood (*Dalbergia eremicola*), needlebush (*Glochidion puberum*), persimmon (*Diospyros kaki*), pomegranate (*Punica granatum*), and soybean (*Glycine max*) [4]. In addition, as documented for many insect herbivores, including scale insects [40,41], the feeding preference of CMBS also has the potential for being altered once the infestation has been successfully established on a novel host. Therefore, the study of feeding preferences and host range is crucial in the prediction and prevention of the expansion of CMBS population.

Naturally occurring CMBS infestation has been reported on American beautyberry (*C. americana*) [4], crapemyrtle, *Hypericum* [42], and spirea (*Spiraea*) [43] in the United States. Wang, et al. (2019) reported four more alternative hosts, including henna (*Lawsonia inermis*), pomegranate (‘Wonderful’), sinicuichi (*Heimia salicifolia* Link), and winged loosestrife (*Lythrum alatum* Pursh), from a host range test in the United States. However, many other horticultural crops of great economic importance, such as *Chaenomeles* and additional species of *Buxus, Diospyros, Malus,* and *Rubus,* have yet to be tested as potential hosts in the United States. In a preliminary test, gravid females were found to develop on ‘Fuji’ apple seedlings when being tied with a CMBS-infested crapemyrtle branch. In addition, as a primary host of CMBS with high market value, pomegranates have been reported to suffer significant level of damage in terms of CMBS infestation in China [44], yet many other common pomegranate cultivars remain untested for their susceptibilities to CMBS.

*Buxus, Chaenomeles, Diospyros, Malus, Punica,* and *Rubus* are valuable fruit or ornamental crops in the United States, as well as other parts of the world. Boxwood, or *Buxus,* is one of the best-selling woody plants in the United States [5]. *Diospyros*, commonly known as persimmon, is the largest genus in Ebenaceae, with more than 500 species of valuable crops with versatile uses with utilizations as edible fruit, timber, ornamental plants, and medicinal usage in east Asia [45,46]. In the United States, *D. virginiana* is known as a native species and is cultivated as a fruit crop [47]. *Malus domestica*, commonly known as apple, is considered one of the most valuable fruit crops in the United States [48]. The United States is one of the largest producers of apple, second to China, in the world [49]. Blackberries and raspberries, commonly referred to as brambles, are also highly valued fruit crops in the genus *Rubus*. In the United States, California, Oregon, and Washington are three of the leading producers of blackberries and raspberries [50].

Finally, *Punica*, or pomegranate, is a species of fruit crop that has been highly valued as food and for medicinal purposes throughout human history [51]. Some cultivars of pomegranates have been developed as ornamental plants [52,53]. Pomegranates are known to be native to central Asia, while years of cultivation has spread their distribution to a wide range of geographical regions including Europe and the United States [51]. According to latest data, the United States National Clonal Germplasm Repository in Davis, California, is currently maintaining 194 accessions of pomegranates, 82 of which were cultivars originated from the United States [54].

Potential pitfalls and challenges in pomegranate production involve damages and diseases related to insects, fungi, and bacteria [51]. Arthropod pests such as the black borer *Apate monachus* [55] and aphids [51], are known to cause major damage on pomegranate production. Scale insects, including CMBS, have been reported to be an issue and caused severe damage in pomegranate production in China [44]. In fact, CMBS is also referred to as pomegranate felt scale in China [56]. The infestations of CMBS were found on different cultivars of pomegranate, with various sizes ranging from container plants [57], to 8–10-year-old pomegranate trees at a nursery [44]. Wang et al. 2019 have confirmed *P. granatum* ‘Wonderful’, one of the most popular cultivars in the United States [58], as a suitable host for CMBS, however, the threat of CMBS to other pomegranates in the U.S. remains undetermined.

Plant species and cultivars within the genus of *Buxus, Diospyros, Malus, Punica,* and *Rubus* have been previously reported as the potential alternative hosts of CMBS outside of the United States [4,59,60,61], which leads to concerns for their production and utilization in the United States, as well as other regions of the world. Therefore, it was hypothesized that plant species from these genera are potentially susceptible to CMBS attack, and thus selected for evaluation in CMBS feeding trails. Three independent feeding studies were designed to investigate the ability of CMBS to infest plant species within selected genera, along with its feeding preference.

In 2016 and 2019, two CMBS feeding trials were conducted to evaluate CMBS preference among 35 available pomegranate cultivars in the United States. Another feeding study was conducted in 2019 to evaluate the host preference of CMBS on plant species and cultivars within seven genera. In summary, a total of 49 plant species or cultivars from *Buxus, Chaenomeles, Diospyros, Heimia, Malus, Punica,* and *Rubus* were evaluated for their susceptibility to CMBS. This study aims to expand the current knowledge on the host range of CMBS in the United States, and help the development of an effective integrated pest management program for controlling this pest insect.

## 2. Materials and Methods

### 2.1. Insect Source and Plant Material

Insect samples used in three feeding studies were collected prior to inoculation from June to September in 2016, and from May to June in 2019. The branches/twigs infested with CMBS were collected from crapemyrtle trees on campus (Texas A&M University, College Station, TX), and stored in zip-lock bags at room temperature (25 °C). The CMBS collected from infested plants were used for the experiments immediately, or within one or two days after the collection.

For pomegranate feeding preference trials, 30 cultivars were obtained from University of Florida and Texas A&M AgriLife Research & Extension Center at Uvalde, TX, and evaluated in 2016 (Table 1). In 2019, cuttings from 14 pomegranate (*P. granatum*) cultivars (collected from Texas A&M AgriLife Research & Extension Center, Uvalde, TX, USA) were rooted in tree tubes and transplanted in containers (3.79 L) for the CMBS feeding preference trial (Table 1).

In 2019, a separate host range confirmation study was conducted to evaluate the feeding preference of CMBS on seven plant genera. Candidate plant species were selected from previously reported CMBS hosts outside of the United States. Commercially available crapemyrtle hybrid cultivar *Lagerstroemia* ‘Spiced Plum’, which could manifest a typical heavy infestation when under CMBS attack, was thus utilized as a control. Fourteen plant species and cultivars from seven genera, including *Buxus harlandii, B. microphylla* var. *koreana* × *B. sempervirens* ‘Green Gem’, *Chaenomeles speciosa* ‘Texas Scarlet’, *Diospyros rhombifolia, D. virginiana, Heimia salicifolia, L.* ‘Spiced Plum’, *Malus angustifolia, M. domestica* ‘Fuji’, *M. domestica* ‘Red Delicious’, *Rubus* ‘Arapaho’, *R. fruticosus* ‘Prime Ark Freedom’, *R. idaeus* ‘Dorman Red’, and *R.* ‘Navaho’, grown in containers (3.79 L), were used in the experiment (Table 2).

### 2.2. Pomegranate Feeding Trials

Two independent feeding trials were conducted at an outdoor container nursery in 2016 and in a greenhouse (25 ± 5 °C, 50 ± 10% relative humidity) in 2019, respectively, to evaluate the cultivar preferences of CMBS among a total of 35 pomegranate cultivars. Both trials were located on Texas A&M University campus (30°36′31.9″ N, 96°21′01.7″ W).

For the 2016 trial, 30 cultivars with 5 replications were tested (Table 2). Three rounds of inoculations were conducted on 6 June 2016, 22 July 2016, and 30 September in 2016, respectively. For the first and second inoculations, CMBS ovisacs were removed from infested plants and placed onto all test cultivars. For the third inoculation, One CMBS-infested branch (10-cm in length) per plant was used to inoculate all tested cultivars. Crapemyrtle bark scale infestation was recorded around one month after the third inoculation. For the 2019 trial, 14 cultivars with 5 replications were evaluated (Table 2). Two rounds of inoculation using one CMBS infested branch (10 cm in length) were conducted on 15 May 2019 and 25 June 2019, respectively. The number of male pupae and gravid females were recorded on 10 August 2019.

Male pupae and gravid females, recognized as white felt-like coverings or ovisacs, were distinguished by their size and shape. Male pupae were identified as white sacs with an elongated shape (around 1 mm long, and 0.5 wide), while gravid females were identified as white ovisacs with a much larger size (around 2 mm long, and 1 mm wide) and rounded shape [4]. 

### 2.3. Host Range Confirmation on Seven Genera

Plant chambers (127 × 127 × 127 cm) were constructed using ½ inch PVC pipes as a frame, and enclosed by Chiffon mesh netting (Fabric Wholesale Direct, Farmingdale, NY 11735). Zippers were sewn onto the mesh fabric to form openings for watering and moving the plants. All plant chambers were placed inside a greenhouse (25 ± 5 °C, 50 ± 10% relative humidity), located on Texas A&M University campus (30°36′31.9″ N, 96°21′01.7″ W).

Plants were placed inside the plant chambers, before inoculating with CMBS with one set of fourteen plant species and cultivars per chamber. Each set of plants was replicated three times. One infested crapemyrtle branch with all but five fresh ovisacs removed was attached to each test plant, on 13 May and 15 June, 2019, respectively, to ensure the successful inoculation and survival of CMBS on test plants. The numbers of male pupae and gravid females per plant were recorded biweekly from June to September, and monthly from October to December 2019.

### 2.4. Data Analysis and Statistics

Data on the 2016 pomegranate trial was not statistically analyzed, because ovisacs were only found on two plants. In 2019, one-way analysis of variance (ANOVA) was used to analyze the number of male pupae and gravid females found on 14 pomegranate cultivars. Sex ratio (male to female ratio) was calculated using the mean number of male pupae divided by gravid females for each plant species and cultivars. Log transformed of the original data (No. of CMBS +1) was performed to improve normality of data distribution. Data for the number of male pupae and gravid females on seven genera from June to December 2019 were analyzed using multivariate analysis of variance (MANOVA) for repeated measures of all infested plant species and cultivars. Univariate split-plot approach ANOVA was also conducted for the numbers of male pupae and gravid females recorded on all infested plant species and cultivars, and multiple means were compared using All Pair, Tukey HSD with JMP software (JMP Pro14, Statistical Analysis System, Cary, NC, USA).

## 3. Results

### 3.1. Pomegranate Feeding Trials

For the 2016 trial, only one gravid female was found on ‘Angel Red’ and one on ‘Sumbar’. No male pupa was found on any cultivar evaluated. For the 2019 trial, male pupae were found on ‘Double Pink’, ‘Gissarskii Rozovyi’, ‘Kandahar’, ‘Kara Bala Miursal’, ‘Kazake’, ‘Mollar’, ‘San Antonio’, and ‘Sumbar’, and gravid females were found on ‘Double Pink’, ‘Gissarskii Rozovyi’, ‘Kandahar’, ‘Kara Bala Miursal’, ‘Kazake’, ‘Mollar’, ‘Pecos’, ‘San Antonio’, ‘Sogidavna’, ‘Sumbar’, and ‘Surh-Anor’. No sign of CMBS infestation was observed on ‘Big Red’, ‘Kara Kalinski’, and ‘Russian 18’ in 2019 (Figure 1). Although the majority of cultivars included in this trial had CMBS infestations, the number of gravid females were less than 10 on all tested cultivars. The highest number of gravid females were observed for ‘Kandahar’, ‘Kara Bala Miursal’, and ‘Mollar’ (Figure 1). No significant difference in terms of CMBS infestation was found among all test cultivars in 2019.

### 3.2. Host Range Confirmation on Seven Genera

During the 30 weeks (May to December 2019) of the experiment, CMBS male pupae or gravid females were observed on seven out of 14 test plant species (*C. speciosa, D. rhombifolia, H. salicifolia, L.* ‘Spiced Plum’, *M. angustifolia, M. domestica* ‘Fuji’, and *M. domestica* ‘Red Delicious’). Only male pupae and no gravid female were found on *D. rhombifolia* and *M. angustifolia* (Figure 2 and Figure 3). The number of male pupae and gravid females varied significantly across the entire experiment, and the effect of data collection time (Time, *p* < 0.001), plant species and cultivars (Plant species, *p* < 0.001), and the Time by Plant species interactions (*p* < 0.001) were identified for the population dynamics of CMBS. The CMBS populations on all infested plant species and cultivars showed peaks between week 16 to 20 (September 2019) after inoculation (Figure 2 and Figure 3). The number of CMBS was highest on the control plant (*L.* ‘Spiced Plum’) in September (Table 3), and throughout the entire experiment (Table 4), compared to the other infested plant species and cultivars. The number of male pupae on ‘Spiced Plum’ after week 10 showed a significant increase, and remained the highest for the remainder of the experiment (Appendix A). Similarly, the number of gravid females on ‘Spiced Plum’ after week 10 were higher than all the other test plants for the remainder of the experiment (Appendix A).

### 3.3. The Effects of Plant Hosts on Insect Development

No male pupae and gravid females were observed on *B. harlandii, B. microphylla* var. *koreana* × *B. sempervirens, D. virginiana, Rubus* ‘Arapaho’, *R. fruticosus, R. idaeus* ‘Dorman Red’, and *R.* ‘Navaho’, The male pupae were first observed on *D. rhombifolia*, *H. salicifolia*, *L.* ‘Spiced Plum’, and *M. domestica* ‘Fuji’ in week 3. The gravid females were first observed on *L.* ‘Spiced Plum’ in week 3. The delayed onset of infestation was observed for *C. speciosa*, *H. salicifolia, M. domestica* ‘Fuji’, and *M. domestica* ‘Red Delicious’. The first detection of gravid females on *C. speciosa*, *H. salicifolia*, and *M. domestica* ‘Fuji’ was 8 weeks after inoculation (WAI), and 12 WAI on *M. domestica* ‘Red Delicious’, compared to 3 WAI of the first detection on *L.* ‘Spiced Plum’ (Figure 3). The different onset of infestation could be due to the varied development durations of nymphal stages, which shows the different level of adaptive behavior of CMBS towards novel hosts.

The highest mean number of male pupae and gravid females on *L.* ‘Spiced Plum’ was 634.3 ± 206.4 (mean ± SE) male pupae at week 20 and 167.0 ± 52.5 (mean ± SE) gravid females at week 18, respectively. Both male pupae and gravid females were observed on *C. speciosa*, *H. salicifolia, M. domestica* ‘Fuji’, and *M. domestica* ‘Red Delicious’, with lower levels of infestation compared to *Lagerstroemia.* In addition to *L.* ‘Spiced Plum’, the highest mean number of male pupae recorded during the experiment was 3.3 ± 2.0 (mean ± SE) on *C. speciosa*, 28.7 ± 15.9 (mean ± SE) on *H. salicifolia*, 4 ± 2.1 (mean ± SE) on *M. domestica* ‘Fuji’, and 5.3 ± 1.9 (mean ± SE) on *M. domestica* ‘Red Delicious’, respectively. The highest mean number of gravid females recorded during experiment was 3.3 ± 0.9 (mean ± SE) on *C. speciosa*, 10.0 ± 6.4 (mean ± SE) on *H. salicifolia*, 3.7 ± 2.7 (mean ± SE) on *M. domestica* ‘Fuji’, and 1.3 ± 0.3 (mean ± SE) on *M. domestica* ‘Red Delicious’, respectively. Only a few male pupae were observed on *D. rhombifolia* (0.3 at week 3) and *M. angustifolia* (1 at week 14) (Figure 2). The effect of host plant in altering insect sex differentiation was observed in this study, as the male to female ratios on *C. speciosa*, *H. salicifolia, L.* ‘Spiced Plum’, *M. domestica* ‘Fuji’, and *M. domestica* ‘Red Delicious’ was 0.81:1, 2.25:1, 3.67:1, 0.99:1, and 7.23:1 respectively, throughout the experiment.

## 4. Discussion

The pomegranate feeding trials conducted in 2016 and 2019 showed very low levels of CMBS infestation on all tested pomegranate cultivars. Lack of infestation on almost all pomegranate plants could be due to the outdoor conditions. The results from 2019 feeding trial was similar to results from Wang et al. 2019, who showed a low level of infestation (around 10 gravid females found) on ‘Wonderful’ pomegranate. Although CMBS has been reported to cause serious damage to pomegranates in China [44], our results did not indicate a level of damage on the tested cultivars comparable to crapemyrtles. While our result did confirm that pomegranates serve as alternative hosts for CMBS, it is also not uncommon that the severity of damages caused by the same pest insect varies on different genotypes [51].

The geographical origin of the pomegranate cultivars might play a role in determining the susceptibility of host plants to CMBS. According to online database [54,62], the 35 tested pomegranate cultivars in 2016 and 2019 have origins in either Georgia, India, Iran, the Soviet Union or Russia, the United States, or Turkmenistan. No tested cultivar was known to be from China, which suggests that the reported heavy CMBS infestations on pomegranates may be confined to certain pomegranate species or cultivars developed in China.

As an indicator for successful reproduction, gravid female and oviposition are often used to evaluate the performance of an insect on novel hosts [63,64,65]. The varied WAI for the first gravid female found on *C. speciosa*, *H. salicifolia, M. domestica* ‘Fuji’, and *M. domestica* ‘Red Delicious’, suggest that the developmental times from nymph to gravid female were at least twice as long compared to *L.* ‘Spiced Plum’. However, the possibility remains for CMBS to adapt and alter feeding preference on novel hosts after several generations or through mutations, as similar phenomena have been reported on other insect species [66,67,68]. The number of gravid females found on *H. salicifolia* was similar to the number reported by Wang et al., 2019 (less than 50 gravid females), while the sex ratio was unknown previously.

No gravid female was observed on *B. harlandii*, *B. microphylla* var. *koreana* × *B. sempervirens*, *D. rhombifolia*, *D. virginiana*, *M. angustifolia*, *R.* ‘Arapaho’, *R. fruticosus*, *R. idaeus* ‘Dorman Red’, or R. ‘Navaho’ in our current study, which might conclude that these species are not suitable hosts for CMBS. However, previous reports [59,61] reported that several plant species in *Buxus*, *Diospyros*, and *Rubus* were CMBS hosts. Similar phenomenon was observed in our study, where gravid females were found on *M. domestica* (originated from Central Asia) [69], but not on *M. angustifolia* (native to the North America) [70]. This suggests that CMBS have different feeding preference toward plants within the same genus, and higher host acceptance might be found especially on certain plant species or cultivars developed outside of the United States.

Crapemyrtle bark scale infestation differed greatly on different crapemyrtle species and cultivars. According to Wang et al. (2019), the highest number of gravid females on *L. indica* × *fauriei* ‘Natchez’ was higher (482 ± 92) than the highest number found on *L.* ‘Spiced Plum’ in this study. Based on current observations, the highest number of male pupae can reach near 1000 on *L. limii* and *L. subcostata* grown in 3.79 L containers, while, under similar conditions, the number of pupae were found to range from 11 to around 400 on *L. fauriei* ‘Kiowa’, *L. caudata*, *L. indica* ‘Dynamite’, and *L. speciosa*. For gravid females, the highest number was found to be around 600 on *L. limii* grown in 3.79 L containers, while, under similar conditions, the highest number was between 200 and 400 on *L. subcostata* and *L. fauriei* ‘Kiowa’, and the highest number was below 100 on *L. caudata*, *L. indica* ‘Dynamite’, and *L. speciosa* [71]. Thus, we found that the highest number of male pupae and gravid females were 634 and 167, respectively, on *L.* ‘Spiced Plum’ grown in 3.79 L containers, which suggests that the level of possible CMBS infestation of ‘Spiced Plum’ would be considered ‘medium’, among other tested crapemyrtle species and cultivars.

The different infestation among crapemyrtle species and cultivars could be related to several factors, including the predation from nature enemy, or certain plant physical or biochemical properties. In this study, the influence of a natural enemy was mostly eliminated, except for the pomegranate feeding trial conducted in 2016, where the experiment was conducted in an outdoor condition. Interestingly, the 2016 experiment showed very limited presence of CMBS on tested cultivars, which suggests that conducting experiment in either a greenhouse or enclosed plant chamber could be effective in excluding the factor of natural enemies. Without the influence of natural enemies, the factors influencing CMBS population or host selection could be refined to plant properties.

Plant physical characteristics, such as plant size and plant physical structure, are known to contribute to the insect acceptance of host plant [72]. The plant size may influence the development of infestation by increasing population density and selection pressure [73]. As CMBS primarily settle and feed on of stem or trunk of plants, a larger size plant would have bigger surface area to host a higher level of infestation, without creating extra ‘pressure’ for the CMBS population. ‘Natchez’ is a large crapemyrtle tree that grows up to 30 feet in height [74], while ‘Spiced Plum’ is expected to mature at a smaller size [75]. However, this could not explain the different infestation levels exhibited among plants of similar size. For example, ‘Spiced Plum’ (grown in 3.79 L container) had a higher level of infestation compared to a similar size crapemyrtle, ‘Dynamite’ (grown in 3.79 L container), from a different feeding experiment [71].

Crapemyrtle bark scale host selection may also involve the development of insects on hosts with a certain plant structure. According our observation and previous reports, CMBS tend to feed on the cracks and crevice, or the wounded area of plant [2]. This suggests that CMBS might find it harder to accept certain plants with a smooth stem surface structure as a host, such as several *Rubus* species included in this study.

Another major factor that affects CMBS host selection and the level of infestation may be plant biochemical properties, such as the composition of saccharides and amino acids [76], which leads to the variation of population dynamics, such as altered development duration and sex ratio. The altered sex ratio of insect populations was found when the insect feeds on different hosts [77,78], or under environmental factors such as temperature [79]. The sex ratio of CMBS on ‘Spiced Plum’ was 3.67:1 (male:female) in this study (Table 4), whereas the sex ratio on ‘Natchez’ remains unknown from previous studies.

The larger number of male pupae and gravid females on *Lagerstroemia* indicates that CMBS showed less feeding preference on other tested plants. Compared to *Lagerstroemia,* there is lower risk in terms of economic damage for *B. harlandii, B. microphylla* var. *koreana* × *B. sempervirens*, *C. speciosa*, *D. rhombifolia*, *D. virginiana, M. domestica* ‘Fuji’, *M. domestica* ‘Red Delicious’, *M. angustifolia*, *R.* ‘Arapaho’, *R. fruticosus, R. idaeus* ‘Dorman Red’, and *R.* ‘Navaho’, as well as pomegranates, when being exposed to CMBS, since no infestation was recorded, or the recorded infestation might not reach the threshold for aesthetic and economic damages.

Economic thresholds (ET) of plant damages incurred by pest insects are widely used as advisory tools to determine action points for effective treatments [80,81,82]. One study showed that soybean aphid has an ET of 250 insects per plant [82], however, the ET could be lower for CMBS infesting ornamental crops, considering the negative effect on the aesthetic value resulting from defoliation, loss of flowering, and the accumulation of sooty mold [2,4,7].

The ET of CMBS has not yet been well categorized, and could be subject to change according to the specific crops. For example, ornamental crops, such as impatiens [83], tend to have lower tolerance compared to other economically important food crops, such as strawberry [84], when exposed to similar type of pest damage. Therefore, crapemyrtles and *C. speciosa* present in this study might have a low ET, since the presence of small numbers of CMBS could greatly decrease the ornamental values [2].

The infestation of CMBS had showed the level of damage well above ET for the control plant (*L.* ‘Spiced Plum’) in terms of aesthetic value (Appendix A). The infestations found on all alternative hosts in this study should be considered under the ET, due to the low number of insects presented (Appendix A). However, the source of CMBS inoculant in this experimental setting was relatively low, as only 10 CMBS ovisacs were used to inoculate each test plant. Under a natural setting, it is possible that plants infested with CMBS would produce large, and continuous influxes of insects to the nearby healthy plants. Therefore, the effect of heavier CMBS inoculation might lead to different levels of infestations on alternative plant hosts, which could be further investigated in future studies.

## 5. Conclusions

Crapemyrtle bark scale feeding preference studies on eight genera reported infestation on apple (*M. domestica*), *C. speciosa*, *D. rhombifolia*, *H. salicifolia*, *L.* ‘Spiced Plum’, *M. angustifolia*, and pomegranate cultivars including ‘Double Pink’, ‘Gissarskii Rozovyi’, ‘Kandahar’, ‘Kara Bala Miursal’, ‘Kazake’, ‘Mollar’, ‘Pecos’, ‘Angel Red’, ‘San Antonio’, ‘Sogidavna’, ‘Sumbar’, and ‘Surh-Anor’. No sign of CMBS infestation was observed on *R.* ‘Arapaho’, *R.* ‘Navaho’, *R. idaeus* ‘Dorman Red’, *R. fruticosus*, *B. microphylla* var. koreana × *B. sempervirens*, *B. harlandii*, and *D. virginiana*. No sign of CMBS infestation was observed on pomegranate cultivars including ‘Al-Sirin-Nar’, ‘Apseronski Krasnyi’, ‘Austin’, ‘Azadi’, ‘Bala Miursal’, ‘Christina’, ‘Desertnyi’, ‘Elf’, ‘Entek Habi Saveh’, ‘Girkanets’, ‘Gissarskii Rozovyi’, ‘JD’, ‘Kandahar’, ‘Kara Kalinski’, ‘Kazake’, ‘Larkin’, ‘Molla Nepes’, ‘Mollar’, ‘Mridula’, ‘Russian 18’, ‘Salavatsk’, ‘Sirenevyi’, ‘Sogidavna’, ‘Spanish Sweet’, ‘Sweet’, ‘Vkusanyi’, and ‘Wonderful’ in 2016 and 2019.

This study expands the current host range for CMBS and confirmed the abilities of CMBS to infest alternative plant species, especially plants within genera of *Chaenomeles, Heimia, Malus,* and *Punica*, with different feeding preferences. The damage caused by CMBS on alternative plant hosts in this study is relatively low compared to *Lagerstroemia*, however, the effect and potential threats of CMBS to its current list of alternative hosts, as well as other plant species, should be evaluated under more experimental conditions.

## Figures and Tables

**Figure 1 insects-11-00399-f001:**
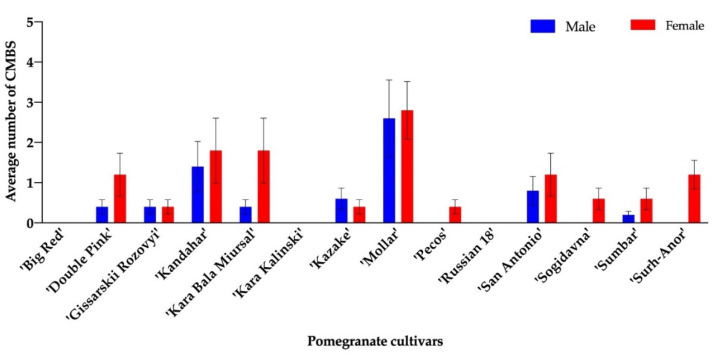
Number of male pupae and gravid females of *Acanthococcus lagerstromiae* per cultivar of pomegranates, including ‘Big Red’, ‘Double Pink’, ‘Gissarskii Rozovyi’, ‘Kandahar’, ‘Kara Bala Miursal’, ‘Kara Kalinski’, ‘Kazake’, ‘Mollar’, ‘Pecos’, ‘Russian 18’, ‘San Antonio’, ‘Sogidavna’, ‘Sumbar’, and ‘Surh-Anor’, recorded on August 10th 2019. No male pupa or gravid female was observed on ‘Big Red’, ‘Kara Kalinski’, or ‘Russian 18’.

**Figure 2 insects-11-00399-f002:**
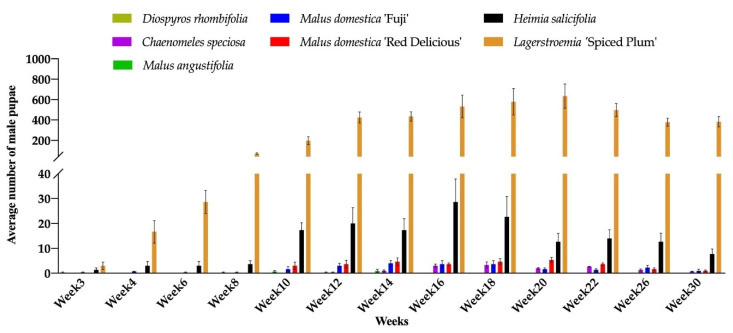
Number of male pupae of *Acanthococcus lagerstromiae* on *Malus domestica* ‘Fuji’, *M. domestica* ‘Red Delicious’, *M. angustifolia*, *Chaenomeles speciosa*, *Diospyros rhombifolia*, *Heimia salicifolia*, and *Lagerstroemia* ‘Spiced Plum’, recorded from June to December 2019.

**Figure 3 insects-11-00399-f003:**
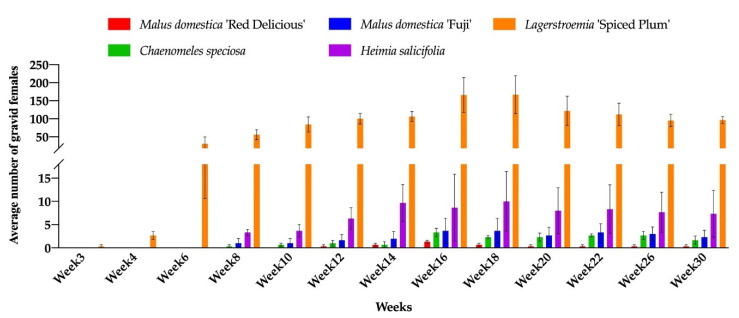
Number of gravid females of *Acanthococcus lagerstromiae* on *Malus domestica* ‘Fuji’, *M. domestica* ‘Red Delicious’, *Chaenomeles speciosa*, *Heimia salicifolia*, and *Lagerstroemia* ‘Spiced Plum’, recorded from June to December 2019.

**Table 1 insects-11-00399-t001:** Pomegranate (*Punica granatum*) cultivars as host candidates of *Acanthococcus lagerstroemiae* used in feeding trials in 2016 and 2019, respectively.

2016		2019
‘Al-Sirin-Nar’	‘Kazake’	‘Big Red’
‘Angel Red’	‘Larkin’	‘Double Pink’
‘Apseronski Krasnyi’	‘Molla Nepes’	‘Gissarskii Rozovyi’
‘Austin’	‘Mollar’	‘Kandahar’
‘Azadi’	‘Mridula’	‘Kara Bala Miursal’
‘Bala Miursal’	‘Russian 18’	‘Kara Kalinski’
‘Christina’	‘Salavatski’	‘Kazake’
‘Desertnyi’	‘Sirenevyi’	‘Mollar’
‘Elf’	‘Sogidavna’	‘Pecos’
‘Entek Habi Saveh’	‘Spanish Sweet’	‘Russian 18’
‘Girkanets’	‘Sumbar’	‘San Antonio’
‘Gissarskii Rozovyi’	‘Surh-Anor’	‘Sogidavna’
‘JD’	‘Sweet’	‘Sumbar’
‘Kandahar’	‘Vkusanyi’	‘Surh-Anor’
‘Kara Kalinski’	‘Wonderful’	

**Table 2 insects-11-00399-t002:** Plant species as host candidates of *Acanthococcus lagerstroemiae* used in feeding preference study in 2019.

Scientific Name	Common Name	Family	USDA Cold Hardiness Zone
*Buxus harlandii*	Harland boxwood	Buxaceae	7–9
*Buxus microphylla* var. koreana × *B. sempervirens* ‘Green Gem’	Boxwood	Buxaceae	7–9
*Chaenomeles speciosa* ‘Texas Scarlet’	Common quince	Rosaceae	4–8
*Diospyros* *rhombifolia*	Diamond-leaf Persimmon	Ebenaceae	7–11
*Diospyros virginiana*	Common persimmon	Ebenaceae	4–9
*Heimia salicifolia*	Sinicuichi	Lythraceae	9–11
*Lagerstroemia* ‘Spiced Plum’	Crapemyrtle	Lythraceae	6–10
*Malus angustifolia*	Southern crabapple	Rosaceae	3–8
*Malus domestica* ‘Fuji’	Apple	Rosaceae	3–8
*Malus domestica* ‘Red Delicious’	Apple	Rosaceae	3–8
*Rubus* ‘Arapaho’	Blackberry	Rosaceae	4–9
*Rubus* ‘Navaho’	Raspberry	Rosaceae	6–10
*Rubus fruticosus* ‘Prime Ark Freedom’	Blackberry	Rosaceae	6–9
*Rubus idaeus* ‘Dorman Red’	Raspberry	Rosaceae	5–9

**Table 3 insects-11-00399-t003:** The number of male pupae and gravid females, and sex ratio (male to female ratio) on five plant species and cultivars infested with *Acanthococcus lagerstromiae* in September 2019.

Plant Species	No. Male Pupae	No. Gravid Females	Sex Ratio
*Lagerstroemia* ‘Spiced Plum’	555.7a ^Z^	166.5a	3.3:1
*Heimia salicifolia*	25.7b	9.3b	2.8:1
*Malus domestica* ‘Red Delicious’	4.2b	1.0b	4.2:1
*M. domestica* ‘Fuji’	3.7b	3.7b	1.0:1
*Chaenomeles speciosa*	3.2b	2.8b	1.1:1

^Z^ Means within each column followed by the same letter are not significantly different according to All Pairs, Tukey Honestly Significant Difference at 0.05 confidence level.

**Table 4 insects-11-00399-t004:** The number of male pupae and gravid females, and sex ratio (male to female ratio) on seven plant species and cultivars infested with *Acanthococcus lagerstromiae* from June to December 2019.

Plant Species	No. Male Pupae	No. Gravid Females	Sex Ratio
*Lagerstroemia* ‘Spiced Plum’	321.5a ^Z^	87.7a	3.7:1
*Heimia salicifolia*	12.6b	5.6b	2.3:1
*Malus domestica* ‘Red Delicious’	2.4b	0.3b	7.2:1
*M. domestica* ‘Fuji’	1.9b	1.9b	1.0:1
*Chaenomeles speciosa*	1.1b	1.4b	0.8:1
*M*. *angustifolia*	0.2b	-	-
*Diospyros* *rhombifolia*	0.03b	-	-

^Z^ Means within each column followed by the same letter are not significantly different according to All Pairs, Tukey Honestly Significant Difference at 0.05 confidence level.

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
