# Peer review of "Feeding Preference of Crapemyrtle Bark Scale (Acanthococcus lagerstroemiae) on Different Species"

_insects, 2020, doi:10.3390/insects11070399_

Round 1

Reviewer 1 Report

The present study represents a contribution in order to protect the native plants against the Crapemyrtle bark scale (CMBS), Acanthococcus lagerstroemiae. It is original research, where it is easy to identify significant efforts, mainly on data collection, which was evaluated in different hosts, covering different agricultural and ornamental interest plants. However, the authors need to address some concerns before it gets the acceptance status. Here below I suggest a major concern that need to be addressed in the manuscript.

Title

-The title must be changed. It is not necessary to use the term ‘eight genera’. I suggest use ‘on different species’ or ‘different hosts’.

Abstract
-The abstract is based on the writing of the manuscript introduction. Does not bring conclusive results of the manuscript.

-Lines 22 and 23 are unnecessary in the abstract.

-Line 34: The keywords need to let the readers an understanding of what the scientific manuscript is about. As a suggestion, I suggest including the term ‘exotic pest’ or ‘invasive pest’.

- Line 33: Apparently, the authors have a great concern with the spread in the USA, however, this can be a problem in other countries, considering it is an invasive species. Therefore, there is no reason to highlight the USA here in the abstract, neither during the Manuscript. It should focus on whether it is a pest that can affect different plants in different countries where the pest does not occur or is expanding.

Introduction
- Firstly, I think is important include the complete scientific name of CMBS considering the authority for a binomial name at least when it is first mentioned, and the date of publication may be specified: Acanthococcus lagerstroemiae Borchsenius, 1960

-The introduction is vague, containing only a paragraph that contextualizes readers with the real purpose of the paper. The introduction is based on economic aspects and pest control but none of this was confirmed in the work. Focus on introducing insect host selection as in lines 76-80 and conclusion in lines 376-381. When examples are needed, please consider coccidian species. The introduction and discussion sections need to be rewritten in a more concise and clear way as well as placed on lines 32 and 33. In both of them, the authors need to adjust the text to follow a coherent flow.

-Lines 61-68: I suggest rephrasing the writing between lines 61-68. The authors did not evaluate any control alternative for the pest. It is not necessary.

-Lines 76-108: Paragraphs longer than six lines make reading difficult.

-Lines 80-86: If the authors really think it is necessary to maintain examples of host selection, I suggest change the examples of plant specialization to Coccidae species.

-Lines 109-131: Economic aspect and production data are unnecessary in this manuscript.

-Lines 140-149: This paragraph needs to be rewritten. What are the questions of the study? What are the hypotheses? The authors need to explain better their aims in this study.

Material and methods section

-The authors used 2016 events in ‘Methods’ section. It is extensive and repetitive. This method no has a significant result to be used.

-Lines 162-169: Please, it is very confused to understand the genera distributed between the species. It is necessary to separate the genera followed by their respective species as in table 2. The authors use a common name in table 1, and a scientific name in Methods section. Please, put the table in sequence in the text.

-Line 164: the authors do not explain why the ‘Spiced Plum’ was utilized as a control. Does this hybrid cultivar have a specific resistance or susceptibility characteristic to A. lagerstroemiae?

-Line 186: the authors do not explain how they observe the male and female difference.  Have the non-gravid females been evaluated?

-Lines 201-208: Did the authors check all ANOVA assumptions (e.g. normality, homoscedasticity, and independence of the errors) before performing the analysis? Please clarify.

Results

-The results should be better clarified. There were some unexplained points. The USDA Cold Hardinness Zone in table 2 does not generate any results.  I suggest the authors spend more time in figure preparations. The figure 4 is not cited in the results topic and was used only in the last paragraph on discussion. I suggest placing it as a supplemental figure since it is not related to results. Furthermore, the image quality is very low. As it is not about the results collected and the quality of the images, I suggest placing Figure 4 as a supplement.

-Lines 229-230: scientific names have already been cited in the text. Use abbreviations. The same in line 260. The same in line 300 and 306-308.

-Lines 232-233: I suggest the statistical analyze in difference in time and plant species. The Wilk’s Lambda and P value are usually for repeated measures.

-Line 234: Change “around’’ for ‘’between 16 to 20 weeks’’. Correspondent a September. 

-Lines 243-245: Change the scientific names to italic.

-Lines 247-249: Why only Malus domestica 'Fuji', M. domestica 'Red Delicious', Chaenomeles speciosa, Heimia salicifolia, and Lagerstroemia 'Spiced Plum' were evaluated to gravid female?

-Line 249: Were the data collected from June to December 2019 or May to December 2019? Explain that better in results section. Why did the authors use June to December?

-Lines 270-279: Add the statistical analyze.

Discussion

-The discussion is very weak in terms of the host selection. In their actual stages, the discussion just compares the results from other authors. In addition, the discussion does not point the possible mechanism involved the development of the host in different plant species or host development considering time, or seasonal period as a factor. The discussion is based on presenting the results and again uses economic aspects, injuries and damages to justify the importance of the study.

-Line 288: The authors did not evaluate any aspect of level of damage.

-Lines 290-292: in fact, the development of CMBS was low. I suggest that the authors do not use the term as ‘prove’.

-Line 293: The authors did not explore susceptibility of host plants to CMBS in manuscript. Only the development in terms of number of gravid female and male pupa was evaluated. It must be careful with the term throughout the text.

-Line 330: The authors use in the discussion the term “physical characteristics” for explaining the differences infestation between species and cultivars. However, more details should be addressed for better understanding of the physical characteristics between plant species. For example, do some of the varieties have physical structures that can provide shelter for the insect, or protection against natural enemies? Do some of these species-varieties have stimulating compounds for food or food supplements, such as exudates from the plant, that guarantee a better development of the species in one plant to the detriment of another?

-In general terms, the discussion will need to be rewritten in a more concise and clear way. Paragraphs longer than six lines make reading difficult.  The authors did not explore any control strategies in this research or economics thresholds to use extensively in the discussion.

Figures

-I believe it is necessary present the Standard bars in figures. This is possible using a ‘Y’ axis in negative scale. The authors can observe that the variance is high and because of that the Standard bars were ‘cut’ in the figure.

-Moreover, observe that there are differences in standardization between the figures. Apparently, the font size differs between the Figures 2 and 3. Observe that in the Figure 1, the legend is placed on the top corner while in the Figures 2 and 3 is at the bottom of the Figures.

-I suggest the authors spend more time in figure preparations. This does not compromise data quality but, helps readers better understand results in a more didactic and quick way.

Reviewer 2 Report

Review of MS with the title "Feeding Preference of Crapemyrtle Bark Scale (Acanthococcus lagerstroemiae) on Eight Genera"

The work presented by the authors deals with the study of a pest often feared and also spreading in different areas of the world. The authors therefore helped to provide more information on possible potential hosts of the pest herbivore. The overall results show that the pest is mainly related to the species Lagerstroemia spp.

In general, I express a positive judgment because the provided framework is sufficiently comprehensive and well written. In addition, the authors present a two-year study that confirming the importance and the need for this type of scientific result also in order to clearly provide information the economic importance of a pest in the near future. A question that I raise to the authors and which they did not indicate how they controlled possible effects of natural enemies naturally or accidentally present in the test area.

Some clarifications on several aspects are required:

Line 70-71. The sentence is not very clear in this context it is advisable to correct or modify it.

Line 76-88. The authors provide a clear picture of several bindings and their relations with hosts and would also be appropriate to mention species that fortunately have not extended their range of host, for example add https://doi.org/10.4081/jear.2020.8879. 

Line 133. For pomegranate pest it is also important to mention at least one other reference example https://www.agriculturejournals.cz/publicFiles/70880.pdf.

Line 155. Provide more information on growing plants. Was the temperature constant throughout the period? The light
used was natural or artificial.

Line 172. Report in Tab 2 a column relating to the botanical family of the species chosen in the test.

Line 172. Give some information on the USDA Cold Hardiness Zone column, why it was introduced ?

Line 204-205. I ask the authors why they preferred to analyze pupae and gravid female inseme together through the adoption of Manova test.

Line 251, 255. Table 3, 4 shows the sex ratio at different plants species. It is appropriate for authors to report this type of adopted test to M&M.

Line 323, 326. Please report gallon in L.

Line 353 -354. The example reported in the reference is not comparable for the type of damage with the species investigated by the authors and therefore I recommend changing the sentence.

Round 2

Reviewer 1 Report

Considering the improvements made after the major review, the manuscript is suitable to be accepted for publication.